# Surgical Management and Reconstruction of Diaphragm, Pericardium and Chest Wall in Mesothelioma Surgery: A Review

**DOI:** 10.3390/jcm10112330

**Published:** 2021-05-26

**Authors:** Pietro Bertoglio, Elena Garelli, Jury Brandolini, Kenji Kawamukai, Filippo Antonacci, Sara Ricciardi, Alessandro Cipolli, Barbara Bonfanti, Sergio Nicola Forti Parri, Niccolò Daddi, Giampiero Dolci, Piergiorgio Solli

**Affiliations:** Division of Thoracic Surgery, IRCCS Azienda Ospedaliera Universitaria di Bologna, 40138 Bologna, Italy; elena.garelli@ausl.bologna.it (E.G.); jury.brandolini@ausl.bologna.it (J.B.); kenji.kawamukai@ausl.bologna.it (K.K.); filippo.antonacci@aosp.bo.it (F.A.); Sara.ricciardi@ausl.bologna.it (S.R.); alessandro.cipolli@aosp.bo.it (A.C.); barbara.bonfanti@ausl.bologna.it (B.B.); sergionicola.fortiparri@ausl.bologna.it (S.N.F.P.); Niccolo.daddi@aosp.bo.it (N.D.); giampiero.dolci@aosp.bo.it (G.D.); piergiorgio.solli@ausl.bologna.it (P.S.)

**Keywords:** malignant pleural mesothelioma, surgery, diaphragm, pericardium, chest wall

## Abstract

Mesothelioma is an aggressive disease arising from parietal pleura. Surgery is a valuable option in the frame of a multimodality treatment. Several surgical approaches have been standardized with the aim of a macroscopic complete resection; these often require homolateral diaphragm and pericardial resection and reconstruction. Extrapleural pneumonectomy (EPP) and extended pleurectomy decortication (EPD) have been recognized as radical surgical procedures. Nevertheless, both operations are technically challenging and associated with a significant rate of peri-operative morbidity and non-negligible mortality. The diaphragmatic and pericardial reconstruction technique is mandatory to avoid respiratory impairment and to reduce post-operative complications like gastric and cardiac herniation. Moreover, in the case of localized chest wall recurrence, surgery might be considered a valuable therapeutical option for highly selected and fit patients. All the technical aspects of the resection and reconstruction of the diaphragm, pericardium, and chest wall are described as well as the possible use of new minimally invasive techniques. In addition, the choice of different prosthetic materials, considering the most recent innovations in the field, are discussed.

## 1. Introduction

Malignant pleural mesothelioma (MPM) is a rare, aggressive cancer arising from parietal pleura. Recognized risk factors include asbestos exposure and viral infection (such as Simian Virus 40, SV40). Different histological subtypes are known: the most frequent are epithelioid, sarcomatoid, and biphasic [1]. The long-term outcomes of MPM are still unsatisfactory, and they are strongly affected by histology; for instance, epithelioid histology is characterized by a less aggressive behavior compared to other histotypes. The current standard treatment of MPM is based on systemic chemotherapy that accounts for platin and pemetrexed [2], but a recent breakthrough of immunotherapy might offer new solutions for the future [3].

Surgery plays different roles in different phases in the management of mesothelioma patients: in the diagnosis, as a radical treatment, and for palliative purposes [1,4].

Radical surgery is usually offered in the frame of a multidisciplinary approach, and chemotherapy is offered either before or after the surgical procedure [5,6,7]. Epithelioid and selected biphasic tumors are ideal candidates, while pure sarcomatoid tumors are usually not indicated for surgery. Early clinical stages (such as clinical N0) are usually suitable for surgery, while advanced diseases are seldom indicated for surgical resection or might be indicated after induction therapies.

## 2. Surgical Procedures for MPM Resection

Given the peculiar anatomical features of the pleura, the correct radical excision of MPM is not achievable because a free margin would consider vital structures such as the aorta, heart, esophagus, and trachea, as well as the whole homolateral chest wall. The aim of surgery is therefore a macroscopic complete resection (MCR) [8], which entails the resection of all the macroscopic disease. In more detail, two different procedures have been validated for the radical treatment of MPM: extrapleural pneumonectomy (EPP) and extended pleurectomy and decortication (EPD) [9]. Both of the procedures encompass a complete resection of the parietal pleura, homolateral diaphragm, and pericardium; while EPP requires a complete pneumonectomy, EPD is a lung-preserving procedure. EPD has gradually become more popular thanks to better long-term outcomes and the better preservation of performance status [1]. Recurrences after MCR are common, accounting for more than 50% of patients in the majority of the reported studies; EPP is usually plagued by distant metastasis, while EPD patients generally develop local recurrences [10]. Nevertheless, data have not been consistent, and the MARS trial, which highlighted a possible detrimental effect of EPP [11], has been harshly criticized by the scientific community [12]. The MARS 2 trial, focusing on the role of EPD, has finished accruing patients, but results are not yet available.

Both EPP and EPD include the routinary resection of the homolateral diaphragm and pericardium, which need to be reconstructed to avoid complications. Moreover, in case of chest wall invasion or focal chest wall MPM recurrence, resection and reconstruction might be required to achieve the MPR.

## 3. Diaphragm

### 3.1. Diaphragm Management

As discussed before, EPP and EPD require diaphragm resection. In a retrospective analysis of their institutional cohort, Sharkey and her colleagues [13] reported outcomes of more than 200 patients focusing on the pathological results and the surgical management of the diaphragm. In their conclusions, the authors stressed the importance of diaphragmatic resection, stating that the risk associated with a routinary phrenectomy is lower than the risk of performing a possible R2 resection.

### 3.2. Types of Prosthesis

No clear and consistent evidence or guidelines are currently available to choose the best material or technique to be used for diaphragm reconstruction. Consequently, the choice of material is usually based on surgeons’ preferences and habits.

Both autologous and alloplastic materials are available. Among alloplastic materials, 2 mm-thick expanded polytetrafluoroethylene (e-PTFE) is probably the most popular among thoracic surgeons for diaphragmatic reconstruction after either EPP or EPD. Several studies have shown a possible integration by the host tissue after several months with fibroblastic and epithelial cell ingrowth [14]. On the other hand, e-PTFE lacks elasticity and could shrink, thus producing a larger amount of scar tissue than other materials [15]. E-PTFE prosthesis are also available and often used in their dual mesh formulation, which is characterized by two different surfaces with the aim of reducing the adhesion of abdominal organs and facilitating the growth of host cells in the thoracic part.

Moreover, meshes made by polyester or polypropylene are also available, though they seldom used for diaphragmatic substitution. Recently, Rolli and colleagues [16] reported the use of a novel non-absorbable synthetic (polyester) mesh with a higher permeability that allowed for fluid exchange and cell migration. In their first 12 patients, the authors reported satisfactory results in terms of resistance and postoperative complications.

The choice of a synthetic alloplastic material allows for a gain in resistance, but, at the same time, it is associated with a non-negligible risk of infection (2.4%) and a herniation risk that may vary from 3.8% to 12% [17].

Conversely, absorbable meshes might reduce the risk of infection and late displacement of the prosthesis that is eventually replaced by host tissue. Biologic meshes are produced by eliminating the immunogenic properties of the allograft so that the scaffold of the basal membrane can be easily re-colonized by the host epithelium. The extracellular matrix is then degraded to enable the full incorporation of the graft [18]. While this process is completed, the alloplastic material ensures the integrity of the reconstruction and provides strength. There are several options regarding this kind of prosthesis: acellular porcine collagen, acellular human dermis, bovine pericardium, and composite meshes (made by a layer of polypropylene and a layer of absorbable mesh).

In their study, Sharkey [13] reported no complications (either displacement or infection) in patients treated with a biological mesh, which had better outcomes than a non-absorbable mesh, but the authors admitted that the low number of patients might have influenced the results.

Though not commonly used, autologous options have been described. These options encompass the use of several types of pedicled muscle flaps to reconstruct diaphragmatic defects: a flap from the external oblique, the transverse rectus abdominis, and the latissimus dorsi flap. Autologous solutions significantly reduce the risk of infection, but they require a longer operative time and have potential complications due to the harvesting procedure. Bedini and colleagues [19] reported the use of the reverse latissimus dorsi flap as an autologous prosthesis after EPP in nine patients with no early or late complications.

### 3.3. Technical Aspects

As reported elsewhere [15], different techniques to anchor a diaphragmatic prosthesis are available according to surgeon preference. Interrupted, non-absorbable stiches are commonly used.

The suturing of the mediastinal side requires attention in order to properly fix the mesh while avoiding abdominal organ herniation and the injury of the mediastinal organs and structures, both on right side (inferior vena cava) and on the left side (aorta and esophagus); as a matter of fact, the left posterior mediastinum is the area where several authors have described the highest incidence of mesh dehiscence, with a percentage of up to 7% [13,15].

Most authors suggest leaving a small rim (maximum 2 cm; Figure 1) of autologous diaphragm along the aortic and esophageal hiatus in order to preserve a safe anchoring space for the diaphragmatic prosthesis (Figure 2) [17]. Reviewing their large experience of almost 500 patients, Sugarbaker and colleagues [17] appreciated that leaving this small rim of diaphragm allowed for a significant reduction of gastric herniation; at the same time, they acknowledged that there are no data regarding possible influence on oncological outcome. As an alternative, Schiavon and coworkers [20] reported the use of an L-shaped titanium plate fixed on the 9th and 10th vertebral bodies, as well as on the posterior arch of the corresponding rib; this titanium plate could be used as an anchor site for diaphragm mesh stiches (Figure 3). The authors reported results after the first 15 cases (nine on the left side), concluding that this technique is safe, does not require significant additional operative time, and allows for the satisfactory anchoring of diaphragmatic mesh with no reported dehiscence.

On the right side, the herniation of the abdominal organ is less frequent due to the presence of the liver [15]. Nevertheless, attention should be paid to avoiding IVC stenosis: as described for the left side, a short rim of autologous diaphragm can be left in place or the diaphragmatic mesh might be tightened to the pericardium edge or the pericardium prosthesis if a pericardiotomy is carried out.

## 4. Pericardium

### 4.1. Pericardium Management

After resection, the pericardium requires a careful reconstruction, with the aim to avoid either cardiac herniation or constrictive pericarditis [21] and their possible fatal consequences. Herniation usually occurs in the first postoperative days, but it can occasionally happen up to six months later [22]. Though left-sided procedures are less prone to developing complications, reconstruction should be carried out on both sides. Sugarbaker and colleagues [17] reported a dramatic reduction of constrictive symptoms after the routinary reconstruction of the pericardium. Concurrently, careful attention should be paid to avoiding cardiac tamponade.

### 4.2. Types of Prosthesis

Similar to diaphragmatic repair, different types of pericardial patches are available. Synthetic meshes are preferred over non-synthetic meshes for the reconstruction of the pericardium; non-absorbable meshes are usually divided into permeable and non-permeable groups. In the non-permeable group, e-PTFE (0.1 mm thickness) is probably the most popular among thoracic surgeons, while in the permeable group, polyester and polypropylene meshes are the most used. The paramount importance of permeability in pericardial meshes is related to the aforementioned risk of cardiac tamponade. Conversely, the infection risk is generally low. Absorbable materials are also used for the reconstruction of the pericardium. The substitution of the pericardium with polyglactin and polyglycolic acid meshes had been reported, but no clear data regarding their strength and absorption time are available. Additionally, bovine patches have been reported for pericardial reconstruction.

### 4.3. Technical Aspects

Pericardial reconstruction should allow a normal cardiac function, avoiding tamponade or a too tight suture that might impair diastolic function.

The patch is usually sutured using interrupted, non-absorbable sutures starting from the posterior part, which tends to be deeper and more difficult to suture [17]. Moreover, on the inferior side, the prosthesis could be sutured to the diaphragmatic patch in order to increase the pericardial space. The fenestration of the mesh allows for the correct outflow of the pericardial fluid towards the pleural space, thus avoiding tamponade.

## 5. Chest Wall

### 5.1. Chest Wall Management

Chest wall resection is not a standard procedure during EPP or EPD. Nevertheless, in some cases, undiagnosed chest wall invasion might require a chest wall resection during radical surgery for MPM. Moreover, in case of local recurrent disease, chest wall resection might be considered in multimodality therapy [23,24,25,26].

### 5.2. Types of Prosthesis

Chest wall reconstruction can be carried out with an absorbable or non-absorbable material, which is then covered with muscle flaps [27]. The choice or association of different materials is related to the extension of the chest wall defect and the surgeons’ choice and habits. Prosthesis is usually required if resection is carried out in the lateral or inferoanterior part of the rib cage.

The available materials are similar to those used for diaphragmatic reconstruction. Flexible meshes can be both absorbable (either synthetic, like polyglactin, or bioprosthetic like cadaveric human dermis, porcine dermis, porcine small intestine submucosa, bovine dermis, and bovine pericardium) and non-absorbable (E-PTFE, polypropylene, and nylon); their flexibility allows for the easier manipulation and uniform distribution of tension. As discussed before, these different materials also have different permeability features, which should be considered by the surgeon while choosing the type of mesh.

Among non-flexible prosthesis materials, methyl methacrylate is the most popular among surgeons. Methyl methacrylate is a resin that can be modelled and tailored according to the patient’s needs in order to correctly fit the defect; it is then wrapped in a non-absorbable mesh (usually polypropylene mesh) and sutured to the chest wall. Despite higher chest wall stability, higher resistance, and lower costs, methyl methacrylate prostheses have been associated with higher rates of wound complications, such as seromas and infections, requiring removal in up to 5% of patients [28].

Lastly, osteosynthesis systems are available. They encompass titanium bars, and they are generally combined with meshes or myocutaneous flaps. Compared to methyl methacrylate, they offer less rigid and more physiologic movement of the chest wall, and they are less prone to infection. However, in a retrospective study, a considerable percentage of these implants failed at long-term follow up due to being either broken or displaced [29].

According to patients’ anatomy and surgical preferences, myocutaneous flaps, pedicled muscle flaps (latissimus dorsi, pectoralis major, rectus abdominis), or, in some cases, even the omentum flap could be used to cover chest wall defects [30].

### 5.3. Technical Aspects

The goal of chest wall prosthesis is to create a rigid surface to avoid both lung herniation and paradoxical movements, as well as to protect the inner organs. Both running and interrupted sutures could be used to tighten a prosthesis, but interrupted stiches allow for the regulation of the tension and position of the prosthesis.

## 6. Role of Minimally Invasive Surgery

The role of video-assisted thoracic surgery (VATS) in the management of advanced MPM was explored in the MesoVATS [31] trial. Nonetheless, to date, minimally invasive surgery still plays a minor role in radical surgery for malignant pleural mesothelioma. In the literature, only few cases of VATS EPP [32,33] have been reported so far. EPP and EPD are complex and extensive procedures. Even when sub-steps can be easily performed by using minimally invasive techniques, thoracotomy is still the currently preferred approach by surgeons in order to achieve satisfactory and correct results in terms of MCR. With improvements of surgical and robotic techniques, this gold standard might change in the near future.

## 7. Conclusions

Diaphragm and pericardial resection and reconstruction are well-defined steps of both EPP and EPD that should allow for a macroscopic, complete resection. For reconstruction, different materials and techniques are now available to maximize the strength of the prosthesis and to reduce complication rates. Chest wall resection might be required in highly selected patients affected by MPM or local recurrences. Currently, the role of minimally invasive surgery is still marginal for the radical resection of MPM.

## Authors Contribution

Conceptualization: P.B., E.G., and P.G.S.; funding acquisition: P.B., P.G.S.; methodology: all authors; project administration: all authors; resources: all authors; supervision: all authors; validation: all authors; visualization: all authors; writing—original draft: P.B., E.G., J.B., and P.G.S.; writing—review and editing: all authors. All authors have read and agreed to the published version of the manuscript.

## Figures and Tables

**Figure 1 jcm-10-02330-f001:**
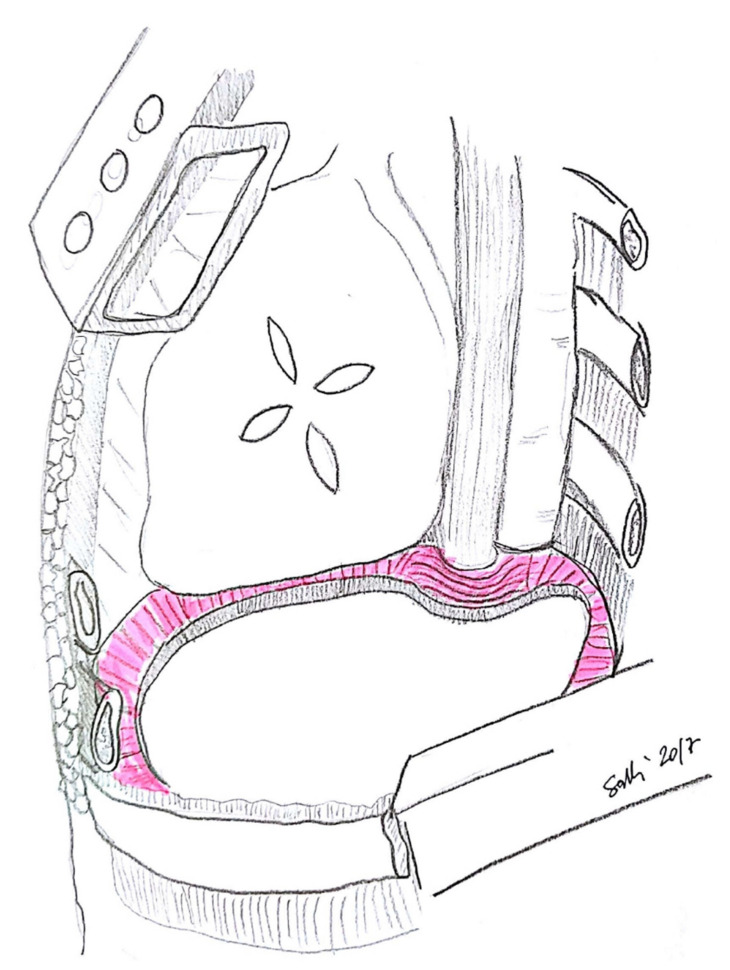
The diaphragmatic rim left in place for suturing diaphragmatic patch (credits: Solli, P.; Brandolini, J.; Pardolesi, A.; Nardini, M.; Lacava, N.; Parri, S.F.; Kawamukai, K.; Bonfanti, B.; Bertolaccini, L. *Diaphragmatic and pericardial reconstruction after surgery for malignant pleural mesothelioma*. J. Thorac. Dis. 2018, 10, S298–S303.).

**Figure 2 jcm-10-02330-f002:**
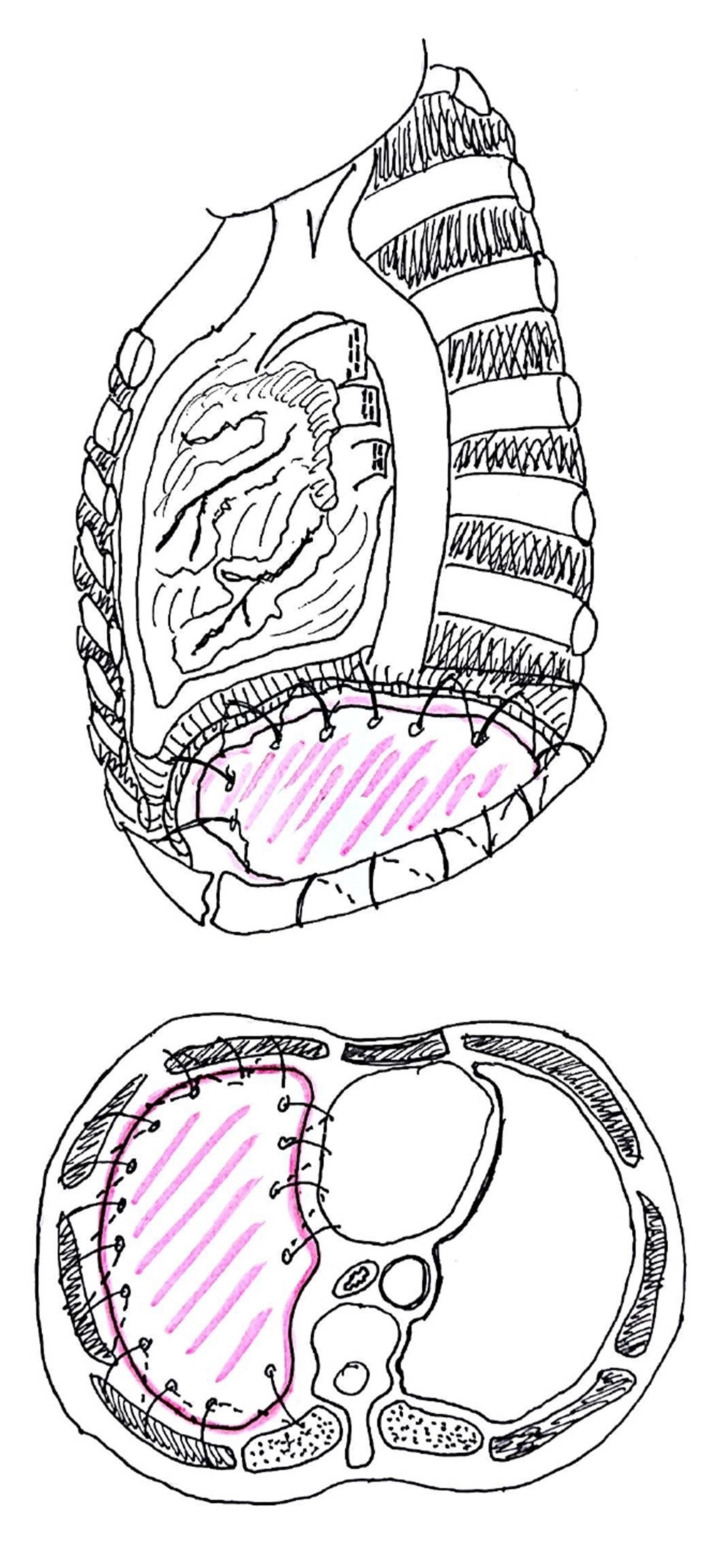
Anchoring of diaphragmatic patch (credits: Solli, P.; Brandolini, J.; Pardolesi, A.; Nardini, M.; Lacava, N.; Parri, S.F.; Kawamukai, K.; Bonfanti, B.; Bertolaccini, L. *Diaphragmatic and pericardial reconstruction after surgery for malignant pleural mesothelioma*. J. Thorac. Dis. 2018, 10, S298–S303.).

**Figure 3 jcm-10-02330-f003:**
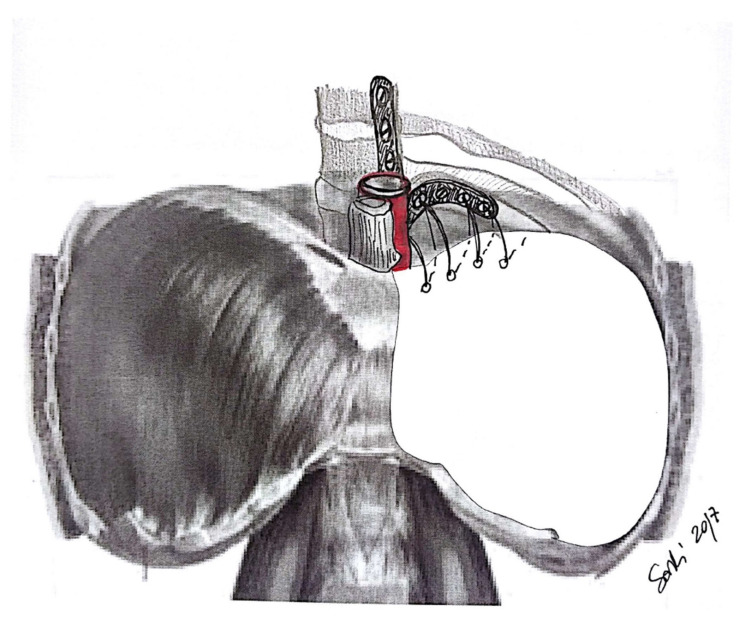
An L-shaped titanium plate used to fix the diaphragmatic patch as described by Schiavon et al. (credits: Solli, P.; Brandolini, J.; Pardolesi, A.; Nardini, M.; Lacava, N.; Parri, S.F.; Kawamukai, K.; Bonfanti, B.; Bertolaccini, L. *Diaphragmatic and pericardial reconstruction after surgery for malignant pleural mesothelioma*. J. Thorac. Dis. 2018, 10, S298–S303.).

## Data Availability

Not applicable.

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
