# Peer review of "Surgical Management and Reconstruction of Diaphragm, Pericardium and Chest Wall in Mesothelioma Surgery: A Review"

_jcm, 2021, doi:10.3390/jcm10112330_

Round 1
Reviewer 1 Report
Dear athors,
congratulations for this review article regarding reconstruction techniques after EPP or EPD. The manuscript is well structured and describes well the different possibilities of reconstruction of diaphragm, pericardium or chest wall after radical surgery for MPM. I would like to point out that besides ePTFE, bovine patches can also be used for reconstruction of both pericardium and diaphragm. Perhaps you could comment briefly on this.
Do you have experience or does the literature describe the temporary use of extrathoracic vacuum therapy after resection of the chest wall and ePTFE reconstruction? Vacuum therapy can be used until final histology is obtained and tumor-free margins are confirmed. Subsequently, definitive wound closure using e.g. muscle flaps would be possible.
Author Response
On behalf of all the coauthors, I would like to thank the reviewer for having read and commented our manuscript. We feel that his/her comments have improved the quality of our work.
The manuscript is well structured and describes well the different possibilities of reconstruction of diaphragm, pericardium or chest wall after radical surgery for MPM. I would like to point out that besides ePTFE, bovine patches can also be used for reconstruction of both pericardium and diaphragm. Perhaps you could comment briefly on this.
As requested by the reviewer, we added a sentence in the pericardial section.
Do you have experience or does the literature describe the temporary use of extrathoracic vacuum therapy after resection of the chest wall and ePTFE reconstruction? Vacuum therapy can be used until final histology is obtained and tumor-free margins are confirmed. Subsequently, definitive wound closure using e.g. muscle flaps would be possible.
We thank the reviewer for this interesting comment. Unfortunately, we do not have any experience with the use of vacuum therapy in this setting and we did not find any experience reported in literature regarding the management of MPM.
Reviewer 2 Report
Dear Authors,
Dear Authors,
Congratulations for this well organised work.
I have few comments for your attention:
- Please add more information in your discussion about the subject of leaving a rim of the diaphragm (about 2 cm) to fix the new diaphragmatic patch
- Please add more information about the rate of recurrence after MCR
- Is there any selective criteria you follow for treating these groups of patients, please add more information about this Task
- Please add more information about the complications in your study period.
Author Response
On behalf of all the coauthors, we would like to thank the reviewer for his/her interesting comments that allowed us to improve the quality of our manuscript.
- Please add more information in your discussion about the subject of leaving a rim of the diaphragm (about 2 cm) to fix the new diaphragmatic patch
We added a sentence for discuss the point raised by the reviewer.
2. Please add more information about the rate of recurrence after MCR
Data regarding recurrences rates after MCR have been added
3. Is there any selective criteria you follow for treating these groups of patients, please add more information about this Task
We collected several published peer-reviewed articles and criteria for enrollement might have changed in different experiences. Nevertheless, we add one sentence in the introduction to better define general criteria for indication to surgery.
4. Please add more information about the complications in your study period.
Although interesting, this review took into account different experiences. Throughout the text we highlighted possible complications due to technical mistakes, but we think that focusing on this particular issue might be out of the main target of our manuscript.